# Bedsharing among breastfeeding physicians: Results of a nationwide survey

**Adetola F. Louis-Jacques**[1]*, **Melissa Bartick**[2,3‡], **Adeola Awomolo**[4‡], **Jiaqi Zhang**[5], **Lori Feldman-Winter**[6], **Stephanie A. Leonard**[5], **Joan Meek**[7], **Katrina B. Mitchell**[8], **Susan Crowe**[5]

1 Department of Obstetrics and Gynecology, University of Florida Health, Gainesville, FL, United States of America, 2 Department of Medicine, Harvard Medical School, Boston, MA, United States of America, 3 Department of Medicine, Mount Auburn Hospital/Beth Israel Lahey Health, Cambridge, MA, United States of America, 4 Department of Obstetrics and Gynecology, University of Arizona College of Medicine, Tucson, AZ, United States of America, 5 Department of Obstetrics and Gynecology, Stanford Medicine School of Medicine, Palo Alto, CA, United States of America, 6 Department of Pediatrics, Cooper Medical School of Rowan University, Cooper University Health Care, Camden, NJ, United States of America, 7 Department of Clinical Sciences, Florida State University College of Medicine, Tallahassee, FL, United States of America, 8 Ridley Tree Cancer Center at Sansum Clinic, Santa Barbara, CA, United States of America

‡ MB and AA contributed equally to this work as second authors.
* louisjacquesa@ufl.edu

**Data Availability Statement:** All relevant data are within the paper and its Supporting Information files.

## Abstract

### Introduction

Bedsharing is common but advised against by the American Academy of Pediatrics. It is unknown if breastfeeding physicians bedshare more or less than the general population.

### Objectives

To determine the prevalence of bedsharing among physicians, their reasons for bedsharing or not, and whether bedsharing was associated with a longer duration of breastfeeding.

### Methods

An online survey was adapted from surveys administered by the Centers for Disease Control and Prevention. The survey was administered to physicians and medical students who birthed children from October 2020 through August 2021. Respondents were asked to report on a singleton birth, and questions centered around sleep practices and breastfeeding. Survival analysis was used to examine the association between bedsharing and breastfeeding duration.

### Results

Of 546 respondents with bedsharing data, 68% reported some history of bedsharing, and 77% were in specialties that involved caring for pregnant people and/or infants. Those who bedshared breastfed an average of four months longer than those who never bedshared (18.08 versus 14.08 months p<0.001). The adjusted risk of breastfeeding cessation was markedly lower for those who bedshared compared to those who did not (Hazard Ratio

**Funding:** The author(s) received no specific funding for this work.

**Competing interests:** The authors have declared that no competing interests exist.

0.57, 95% Confidence Interval 0.45, 0.71). The primary reason for bedsharing was to breastfeed (73%); the primary reason for not bedsharing was safety concerns (92%). Among those who bedshared (n = 373), 52% did not inform their child's healthcare provider.

## Conclusions

Bedsharing is common among our sample of mainly breastfeeding physicians, including those who care for pregnant people and/or infants. It is also associated with a longer duration of breastfeeding, which has implications for population health. Practicing bedsharing implies cognitive dissidence and may affect how physicians counsel about bedsharing. Additionally, lack of disclosure of bedsharing practices has implications for practical guidance about having open non-judgmental conversations and may be a missed opportunity to counsel on bedsharing safety.

## Introduction

Bedsharing has been associated with increased breastfeeding duration and exclusivity [1–3]. Although breastfeeding has been associated with a lower risk of Sudden Infant Death Syndrome (SIDS) [4, 5], the American Academy of Pediatrics (AAP) advises against bedsharing due to concerns that it may increase the risk of sleep-related infant death [6]. However, no clear evidence exists that routine bedsharing itself, in the absence of hazards, causes sleep-related death [4, 7]. The only case control study that examined bedsharing in the absence of risk and used a comparison group of solitary sleeping and room-sharing infants found no significant increased risk [7].

Hazardous circumstances consist of sofa or chair sharing with a sleeping adult, sleeping on soft bedding, sharing a bed with an adult impaired by alcohol or drugs, never having initiated breastfeeding, or sleeping with a preterm or low birthweight infant [8]. Unplanned bedsharing is associated with a two-fold increased risk of SIDS, whereas routine bedsharing without hazards does not appear to be associated with an increased risk [9]. In an effort to avoid bedsharing, parents have fallen asleep with their infants in hazardous locations on sofas or chairs [10].

Populations where bedsharing is common often have low rates of sleep-related infant death [11, 12] associated with low population levels of hazardous risk factors [11]; populations with high rates of death also have high population rates of known hazardous risks (i.e. tobacco exposure) [11].

Bedsharing is common, with 61% of US mothers practicing some bedsharing, regardless of breastfeeding status [13]. Despite the AAP recommendations against bedsharing, first noted in 1997 [14] and widely implemented in 2005 [15], the rates of US bedsharing have not decreased [8, 13]. The US rate of sleep-related infant death has also not decreased in recent years [16], and remains one of the highest rates of all high-income countries [11].

In 2016, Spain's PrevInfad stated that there was not enough evidence to recommend against bedsharing among breastfeeding infants (Level 1 recommendation) [17]. In 2019, the United Kingdom (UK) stopped advising against all bedsharing, except in hazardous circumstances [18]. The UK's National Institute for Health Care and Excellence also concluded there is "no greater risk of harm when parents shared a bed with their baby compared to not bed sharing" [19]. Norway does not advise against bedsharing and advises safe bedsharing [20]. Australia adopted a risk minimization approach in their advice to parents [18, 21], acknowledging that

bedsharing is common and emphasizing avoidance of hazardous circumstances. This position is shared by the Academy of Breastfeeding Medicine (ABM), which states that existing evidence does not support the conclusion that bedsharing causes SIDS in bedsharing breastfeeding dyads [8]. Furthermore, the ABM states that "in the absence of hazardous circumstances, accidental suffocation is extremely rare among bedsharing breastfeeding infants" [8]. Although the AAP guidance in 2022 continues to recommend against bedsharing, it acknowledges that unintentional bedsharing occurs [6].

Breastfeeding physicians may possess medical knowledge about the importance of breastfeeding. Physicians' rigorous work schedules may make breastfeeding challenging and put them at risk for lower duration of any and exclusive breastfeeding [22]. They also may be familiar with recommendations around safe sleep, specifically the recommendations never to bedshare. Some are also expected to convey these recommendations to parents. If physicians themselves are bedsharing, they may find themselves conflicted if they are also expected to counsel parents against it.

It is unknown if breastfeeding physicians bedshare more or less than the general population, given their presumed knowledge about breastfeeding and bedsharing, as well as the demands on their time. We hypothesized that bedsharing was common among physicians and would also be associated with longer breastfeeding duration, consistent with data from the general US population [13]. The aims of this study were to determine the prevalence of bedsharing among physicians, their reasons for bedsharing or not, and whether bedsharing was associated with a longer duration of breastfeeding.

## Methods

### Study instrument and population

An online survey (S1 Appendix) was adapted from surveys administered by the Centers for Disease Control and Prevention Infant Feeding Practices Survey [23], Pregnancy Risk Assessment Survey [24], and investigator-developed questions. For simplicity, we wanted to limit the survey to one birth per respondent. For our specific population, we asked respondents to report on a singleton birth during a self-defined "most strenuous time" in their careers. This period was chosen to capture a period when physicians were most likely to experience reduced breastfeeding duration. We also asked questions regarding their medical training at the time they gave birth.

The recruitment materials invited respondents to participate in a survey about infant "feeding and sleep." A convenience sample were recruited through social media platforms and by email listservs targeting breastfeeding physicians from October 2020 through August 2021. We recruited respondents through the closed Facebook groups, Dr. MILK, which consists exclusively of physicians and medical students who are mothers, as well as Physician Moms Group, which consists of physician mothers. Recruitment invitations were also sent to the Academy of Breastfeeding Medicine listserv, asking for physicians and medical students with a history of birthing a singleton pregnancy. The survey was administered using the Qualtrics platform to physicians and medical students who had given birth. The investigators did not have access to information that could identify individual participants during and after data collection.

Participants were excluded if they were not a physician or medical student, did not have a singleton birth, were missing bedsharing data, or listed "any" breastfeeding data shorter than "exclusive" breastfeeding data. Multiple gestations were excluded since that may confound the probability of breastfeeding continuation. The study was deemed exempt by the Institutional Review Board of Stanford University. The purpose of the study was briefly described via text,

and those who wanted to participate could consent by clicking a link to the survey. The study was unfunded.

We are using terms such as "mothers" "women" and "breastfeeding" when discussing both literature and/or our study respondents. However, we acknowledge that some subjects could have been of any gender identity.

## Outcome measures

Bedsharing was both an exposure and an outcome measure. We considered the respondent to be bedsharing if they answered, "in bed with you" to the question "What did your baby usually sleep in if they slept in the same room as you?" Respondents were also asked if they bedshared "always, sometimes, or never." We also examined bedsharing by time period during the first year of the infant's life: 0–3 months, 4–6 months, and 7–12 months.

We examined breastfeeding outcomes such as duration of any breastfeeding, and whether mothers met their own breastfeeding goals. The breastfeeding duration question was formatted in a way that most of the respondents who were still breastfeeding at the time of the survey left the question unanswered. For the exclusive breastfeeding question, multiple choice responses were provided for duration, with the longest duration being "8 months or greater." Because the survey did not define exclusive breastfeeding, and respondents may have misinterpreted our intended definition, we elected not to use "exclusive breastfeeding" as a separate outcome.

There were 114 respondents whose only breastfeeding data came from their answer to the "exclusive" breastfeeding question. Of these, 66 answered "8 months or greater." The main analysis on breastfeeding duration excluded these respondents. We also performed a sensitivity analysis that included these 114 participants, in which the duration of breastfeeding was assumed to be the reported "exclusive" breastfeeding duration. When the duration of "exclusive" breastfeeding was reported as "8 months or greater," "any" breastfeeding was assumed to be 8 months.

We also examined respondents' reasons for bedsharing or not bedsharing. Respondents could choose as many of the multiple choices presented to them as desired.

## Other variables

We included training status, specialty, marital status, race/ethnicity, geographic location, self-reported postpartum depression, and whether the birth was before 2005 or 2005 or after. We chose this year to reflect the timeframe when AAP strongly recommended against bedsharing and the more widespread adoption of recommendations against bedsharing implemented in the US [15]. We also included a question about whether or not bedsharing was revealed to their child's healthcare provider.

## Statistical analysis

Descriptive statistics, including frequencies and percentages for categorical variables and the mean and standard deviation for quantitative variables, were computed to summarize the participant's characteristics and reasons for bedsharing and not bedsharing. A survival analysis was used to estimate the association between any bedsharing and breastfeeding duration. The survival function for breastfeeding was estimated by the Kaplan-Meier method, and difference of breastfeeding duration between bedsharing was analyzed by the log rank test. Mothers who were still breastfeeding at the time of the survey were censored. For confirming and quantifying the unadjusted and adjusted associations between bedsharing and duration of breastfeeding, we applied univariable and multivariable Cox proportional hazards regression models.

We adjusted for medical specialty, trainee status, race, Hispanic ethnicity, self-reported post-partum depression, and infant birth year before or after 2005. The proportionality assumption was tested by Schoenfeld residual test and scaled Schoenfeld residual plot (S2 Appendix). We used multiple imputation to address missing data among the covariates. Statistical tests were two sided and a p-value of 0.05 was considered significant. Analyses were done with Stata, version 17, College Station, Texas) and R (http://www.r-project.org) software, version 4.1.3.

## Results

There were 806 survey responses. Of our total sample of 546 respondents with bedsharing data (Fig 1 and Table 1), 83% reported themselves as White, 9% as Asian, and 4% as Black/African American. Nearly all respondents initiated breastfeeding (99%) and most indicated they were married or had a domestic partner (98%). Just over half (51%) were attending level, 47% were residents or fellows, and 2% were medical students. Nearly half were pediatricians or medicine-pediatrics (45%), 18% were obstetricians-gynecologists, and 16% were family physicians. Only 10% gave birth before 2005. Nearly all respondents (98%) were from the US or Puerto Rico. The mean duration of breastfeeding was 17 months. Respondents with babies born after 2005 were more likely to report "never bedsharing" (164/488 = 34%) as compared to respondents with babies born prior to 2005 (9/54 = 17%) (p = 0.011). Overall, 68% reported bedsharing (Table 2). Of those respondents who bedshared, 52% did not report it to their child's healthcare provider and 19% did not answer the question.

While the survey showed that both bedsharers and never bedsharers met their breastfeeding goals, the survival analysis revealed that any bedsharing was associated with 4 more months of breastfeeding, 18.08 versus 14.08 months (p<0.001, Fig 2). The proportionality assumption was tested by the Schoenfeld residual test and Schoenfeld residual scaled plot and was not violated (Schoenfeld residual global test p = 0.061). The adjusted Hazard Ratio (HR) for cessation of breastfeeding in bedsharers compared to never bedsharers was 0.56 (95% Confidence Interval [CI] 0.45, 0.71, p<0.001) (Table 3). Even when assuming reported "exclusive breastfeeding" data for those missing any breastfeeding data, those who bedshared breastfed for 3 months longer than those who did not bedshare (adjusted HR 0.56 and 95% [CI] 0.44, 0.70, p<0.001) (Table 4).

Of the respondents who indicated their reasons for bedsharing, 73% did so to breastfeed. In addition, 59% indicated that they bedshared because it helped either the baby or the respondent sleep better. Some respondents indicated that they practiced bedsharing to be close to or bond with baby (28%) or to comfort a fussy baby (45%). Of the respondents who indicated their reasons for not bedsharing, 92% indicated that it was due to safety concerns, and 22% indicated that a doctor or nurse had advised them against sleeping with their baby (Table 5).

## Discussion

We found bedsharing was common among breastfeeding physicians consistent with our hypothesis, including physicians in specialties caring for women/mothers and/or infants. Also consistent with previous literature in the general population and our hypothesis, bedsharing is associated with longer duration of breastfeeding among physicians. There was a greater than 40% HR decrease for breastfeeding cessation.

Those who bedshared did so mainly to breastfeed and to help with sleep. For the overwhelming majority (>90%) of those who did not bedshare, their main reason was a concern for safety, suggesting that they were influenced by medical recommendations. While both groups reported being able to meet their breastfeeding goals, bedsharing physicians breastfed 4 months longer, which is consistent with existing literature [1–3]. This is relevant especially with new US recommendations supporting 2 years of breastfeeding [25]. To our knowledge,

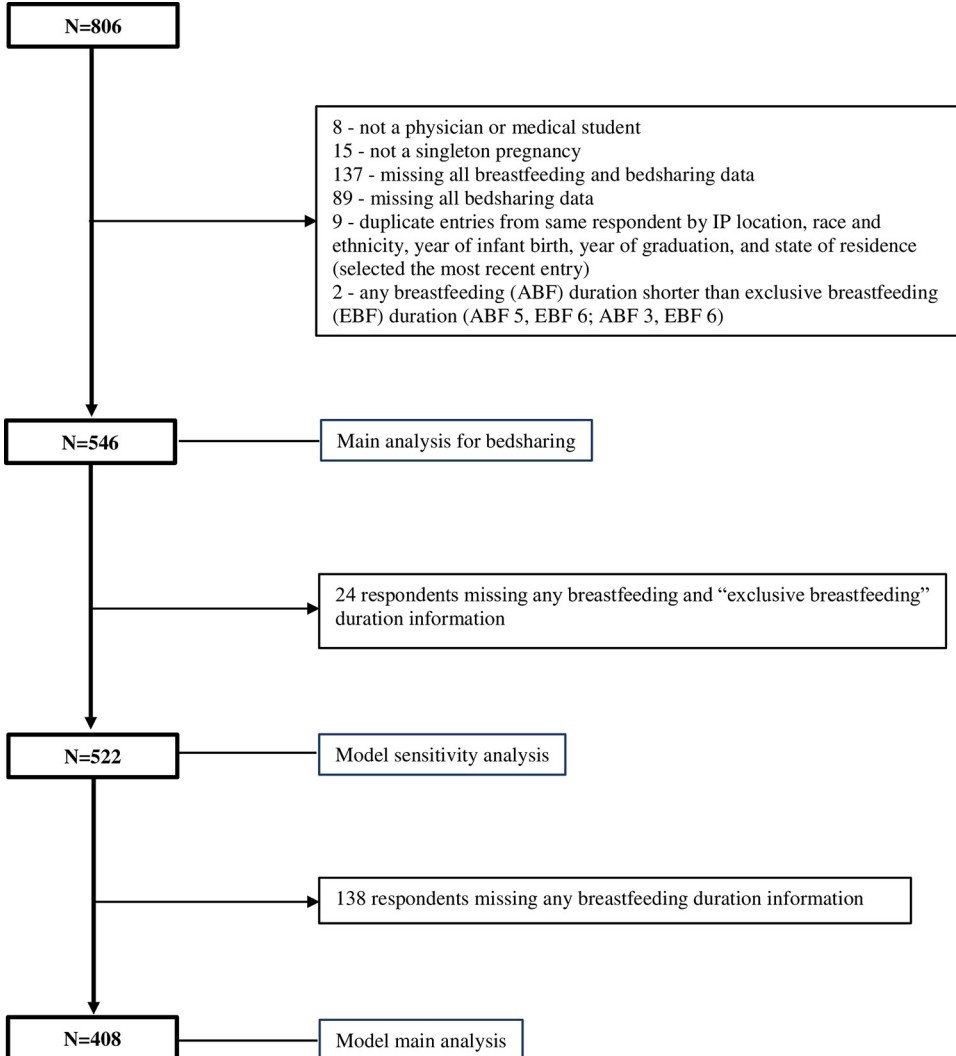

**Fig 1. Participant flow chart.** Inclusion criteria were physicians and medical students who had given birth. Participants were excluded if they were not a physician or medical student, did not have a singleton birth, were missing bedsharing data, or listed "any" breastfeeding data shorter than "exclusive" breastfeeding data. Participants with missing any breastfeeding and "exclusive breastfeeding" duration data were excluded from the model sensitivity analysis. Participants with missing any breastfeeding data were excluded from the model main analysis.

this is the first study conducted about personal experiences of bedsharing among breastfeeding physicians.

In our sample, physicians were weighing safety concerns against the need to facilitate breastfeeding and sleep. It is unclear if intention to bedshare may have played a role in breastfeeding duration and breastfeeding goals, as has been suggested in a previous study [2].

Respondents who did and did not bedshare both met their own breastfeeding goals. This could be that not bedsharing may make breastfeeding more challenging, thereby leading to more modest breastfeeding duration goals. Conversely, it could be that there are additional unmeasured confounders between these groups that may lead to different bedsharing practices and different breastfeeding duration goals. In other words, differences in breastfeeding goals or values between bedsharers and non-bedsharers could explain the difference in breastfeeding duration.

**Table 1. Characteristics of the survey respondents.**

| Characteristics | Respondents with bedsharing data | | | Respondents with bedsharing and breastfeeding data | | |
|---|---|---|---|---|---|---|
| | Overall<br>n = 546[1] | Never bedsharing<br>n = 173[1] | Ever bedsharing<br>n = 373[1] | Overall<br>n = 408[1] | Never bedsharing<br>n = 134[1] | Ever bedsharing<br>n = 274[1] |
| **Race** | | | | | | |
| African American | 22 (4.2%) | 6 (3.5%) | 16 (4.5%) | 21 (5.3%) | 6 (4.5%) | 15 (5.7%) |
| Asian | 46 (8.7%) | 13 (7.6%) | 33 (9.3%) | 33 (8.3%) | 11 (8.2%) | 22 (8.4%) |
| White | 438 (83%) | 150 (88%) | 288 (81%) | 328 (83%) | 115 (86%) | 213 (81%) |
| Other | 21 (4.0%) | 2 (1.2%) | 19 (5.3%) | 14 (3.5%) | 2 (1.5%) | 12 (4.6%) |
| Missing | 19 | 2 | 17 | 12 | | 12 |
| **Hispanic / Latino** | | | | | | |
| Hispanic/Latino | 32 (6.5%) | 8 (5.0%) | 24 (7.2%) | 25 (6.8%) | 8 (6.6%) | 17 (6.9%) |
| Non-Hispanic/Latino | 460 (93%) | 151 (95%) | 309 (93%) | 342 (93%) | 114 (93%) | 228 (93%) |
| Missing | 54 | 14 | 40 | 41 | 12 | 29 |
| **Age of child, in years, mean (SD)** | 6(8) | 5(7) | 7(8) | 8(8) | 7(7) | 9(9) |
| Missing | 4 | | 4 | 3 | | 3 |
| **Marital status at the time of infancy of the child** | | | | | | |
| Married/Domestic Partner | 534 (98%) | 170 (98%) | 364 (98%) | 401 (98%) | 132 (99%) | 269 (98%) |
| Other | 10 (1.8%) | 3 (1.7%) | 7 (1.9%) | 7 (1.7%) | 2 (1.5%) | 5 (1.8%) |
| Missing | 2 | | 2 | | | |
| **On the night that you and your baby laid down together or slept together, who else usually laid down with or slept with you.**\*\* | | | | | | |
| Your husband or partner | 342 (81%) | 47 (77%) | 295 (81%) | 254 (89%) | 39 (81%) | 215 (81%) |
| Your other child or children | 18 (4.2%) | 1 (1.6%) | 17 (4.7%) | 12 (4.2%) | 1 (2.1%) | 11 (4.1%) |
| Other people | 2 (0.5%) | 0 (0%) | 2 (0.5%) | 2 (0.7%) | 0 (0%) | 2 (0.7%) |
| No one else | 91 (21%) | 13 (21%) | 78 (21%) | 66 (23%) | 8 (17%) | 58 (22%) |
| Missing | 121 | 112 | 9 | 93 | 86 | 7 |
| **Postpartum depression** | | | | | | |
| Yes | 137 (25%) | 43 (25%) | 94 (25%) | 109 (27%) | 33 (25%) | 76 (28%) |
| No | 407 (75%) | 129 (75%) | 278 (75%) | 298 (73%) | 100 (75%) | 198 (72%) |
| Missing | 2 | 1 | 1 | 1 | 1 | |
| **Length of maternity leave, mean (SD)** | 9.7(4.5) | 9.8(4.4) | 9.7(4.5) | 9.5(4.5) | 9.6(4.5) | 9.4(4.6) |
| Missing | 10 | 5 | 5 | 8 | 4 | 4 |
| **Trainee Status** | | | | | | |
| Medical student | 11 (2.0%) | 1 (0.6%) | 10 (2.7%) | 8 (2.0%) | 1 (0.8%) | 7 (2.6%) |
| Resident | 196 (36%) | 59 (34%) | 137 (37%) | 153 (38%) | 47 (35%) | 106 (39%) |
| Fellow | 59 (11%) | 13 (7.6%) | 46 (12%) | 48 (12%) | 11 (8.3%) | 37 (14%) |

*(Continued)*

**Table 1.** (Continued)

| Characteristics | Respondents with bedsharing data | | | Respondents with bedsharing and breastfeeding data | | |
|---|---|---|---|---|---|---|
| | Overall<br>n = 546[1] | Never bedsharing<br>n = 173[1] | Ever bedsharing<br>n = 373[1] | Overall<br>n = 408[1] | Never bedsharing<br>n = 134[1] | Ever bedsharing<br>n = 274[1] |
| Not a Trainee | 277 (51%) | 99 (58%) | 178 (48%) | 197 (49%) | 74 (56%) | 123 (45%) |
| Missing | 3 | 1 | 2 | 2 | 1 | 1 |
| **Specialty** | | | | | | |
| Pediatrics/Med-Peds | 242 (45%) | 84 (49%) | 158 (42%) | 197 (49%) | 65 (49%) | 132 (48%) |
| Obstetrics and Gynecology | 95 (18%) | 30 (18%) | 65 (17%) | 79 (19%) | 26 (20%) | 53 (19%) |
| Surgery/Surgical Specialties | 46 (8.5%) | 9 (5.3%) | 37 (9.9%) | 33 (8.1%) | 8 (6.1%) | 25 (9.1%) |
| Internal Medicine/Medicine Specialties | 35 (6.5%) | 12 (7.1%) | 23 (6.2%) | 24 (5.9%) | 7 (5.3%) | 17 (6.2%) |
| Family Medicine | 85 (16%) | 20 (12%) | 65 (17%) | 46 (11%) | 13 (9.8%) | 33 (12%) |
| Emergency Medicine | 16 (3.0%) | 9 (5.3%) | 7 (1.9%) | 11 (2.7%) | 8 (6.1%) | 3 (1.1%) |
| Other* | 23 (4.2%) | 6 (3.5%) | 17 (4.6%) | 16 (3.9%) | 5 (3.8%) | 11 (4.0%) |
| Missing | 4 | 3 | 1 | 2 | 2 | |
| **Geographic Region** | | | | | | |
| West*** | 103 (19%) | 26 (15%) | 77 (21%) | 76 (19%) | 21 (16%) | 55 (20%) |
| Midwest | 128 (24%) | 49 (28%) | 79 (21%) | 95 (23%) | 32 (24%) | 63 (23%) |
| Southwest | 56 (10%) | 17 (9.8%) | 39 (11%) | 41 (10%) | 16 (12%) | 25 (9.2%) |
| Northeast | 111 (20%) | 34 (20%) | 77 (21%) | 79 (19%) | 27 (20%) | 52 (19%) |
| Southeast | 133 (24%) | 44 (25%) | 89 (24%) | 107 (26%) | 35 (26%) | 72 (26%) |
| Puerto Rico | 3 (0.6%) | 2 (1.2%) | 1 (0.3%) | 3 (0.7%) | 2 (1.5%) | 1 (0.4%) |
| Alaska | 10 (1.8%) | 1 (0.6%) | 9 (2.4%) | 6 (1.5%) | 1 (0.7%) | 5 (1.8%) |
| Not in the US | 10 (1.8%) | 1 (0.6%) | 9 (2.4%) | 6 (1.5%) | 1 (0.7%) | 5 (1.8%) |
| Missing | 2 | | 2 | 1 | | 1 |
| **Year of Infant Birth/AAP Sleep Recs** | | | | | | |
| Born before 2005 | 54 (10%) | 9 (5.2%) | 45 (12%) | 54 (13%) | 9 (7%) | 45 (17%) |
| Born 2005 or after | 488 (90%) | 164 (95%) | 324 (88%) | 351 (87%) | 125 (93%) | 226 (83%) |
| Missing | 4 | | 4 | 3 | | 3 |
| **Achieved personal breastfeeding goal** | 413 (78%) | 127 (76%) | 286 (79%) | 299 (73%) | 97 (72%) | 202 (74%) |
| Missing | 16 | 5 | 11 | | | |
| **Exclusive breastfeeding at least 6 mo** | 375 (72%) | 117 (70%) | 258 (72%) | 291 (71%) | 98 (73%) | 193 (70%) |
| Missing | 24 | 7 | 17 | | | |
| **Any breastfeeding at 6 mo** | 41 (10%) | 13 (9.7%) | 28 (10%) | 41 (10%) | 13 (9.7%) | 28 (10%) |
| Missing | 138 | 39 | 99 | | | |

(*Continued*)

**Table 1.** (Continued)

| Characteristics | Respondents with bedsharing data | | | Respondents with bedsharing and breastfeeding data | | |
|---|---|---|---|---|---|---|
| | Overall<br>n = 546[1] | Never bedsharing<br>n = 173[1] | Ever bedsharing<br>n = 373[1] | Overall<br>n = 408[1] | Never bedsharing<br>n = 134[1] | Ever bedsharing<br>n = 274[1] |
| **Any breastfeeding at 12 mo** | 122 (30%) | 48 (36%) | 74 (27%) | 122 (30%) | 48 (36%) | 74 (27%) |
| Missing | 138 | 39 | 99 | | | |
| **Any breastfeeding at 24 mo** | 184 (45%) | 63 (47%) | 121 (44%) | 184 (45%) | 63 (47%) | 121 (44%) |
| Missing | 138 | 39 | 99 | | | |
| **Any breastfeeding longer than 24 mo** | 61 (15%) | 10 (7.5%) | 51 (19%) | 61 (15%) | 10 (7.5%) | 51 (19%) |
| Missing | 138 | 39 | 99 | | | |
| **Duration of any breastfeeding (mo), mean (SD)** | 17(11) | 14(7) | 18(12) | 17(11%) | 14(7%) | 18(12%) |
| Missing | 138 | 39 | 99 | | | |
| **Stopped breastfeeding before survey** | 399 (73%) | 129 (75%) | 270 (72%) | 388 (95%) | 126 (94%) | 262 (96%) |
| **Still breastfeeding at time of survey** | 147 (27%) | 44 (25%) | 103 (28%) | 20 (5%) | 8 (6%) | 12 (4%) |

[1] n (%); Mean (SD)

[2] Pearson's Chi-squared tests; Wilcoxon rank sum test; Fisher's exact test. Fisher's exact test and simulated p-value were applied to geographic region and specialty variable.

% Represents column percentage.

* Other: Physical Medicine and Rehabilitation, Psychiatry, Preventive Medicine, Pathology, Radiation Oncology, Anesthesiology, Interventional Radiology, Radiology.

**Respondents could select more than one response.

***West: include Alaska

mo = Months

The large majority of our sample of mainly US breastfeeding physicians did not follow the AAP safe sleep guidelines on bedsharing, similar to other US populations [13]. In a questionnaire of women in Oregon who had recently given birth, 25.1% reported always bedsharing with their infant, and only 15.7% reported never bedsharing [3]. One would expect physicians to be among the most compliant groups to follow the AAP guidelines, given their type of work in the healthcare sector, and the fact that many are expected to counsel on this guideline as part of their job. This finding demonstrates that the recommendations are not consistent with the lived experiences of the physicians in our sample. We do not know if our respondents were unable or unwilling to follow the guidelines; if they did not believe the guidelines were

**Table 2. Distribution of bedsharing by duration (n = 546).**

| Bedsharing duration (months) | n = 373 | Bedsharing duration | n = 373 |
|---|---|---|---|
| **0–3 only** | 90 (24.1%) | **up to 3** | 90 (24.1%) |
| **0–6** | 68 (18.2%) | **up to 6** | 100 (26.8%) |
| **4–6 only** | 32 (8.6%) | | |
| **4–12** | 33 (8.8%) | **up to 12** | 183 (49.1%) |
| **7–12 only** | 48 (12.9%) | | |
| **0–12** | 98 (26.3%) | | |
| **0–3 and 7–12** | 4 (1.1%) | | |

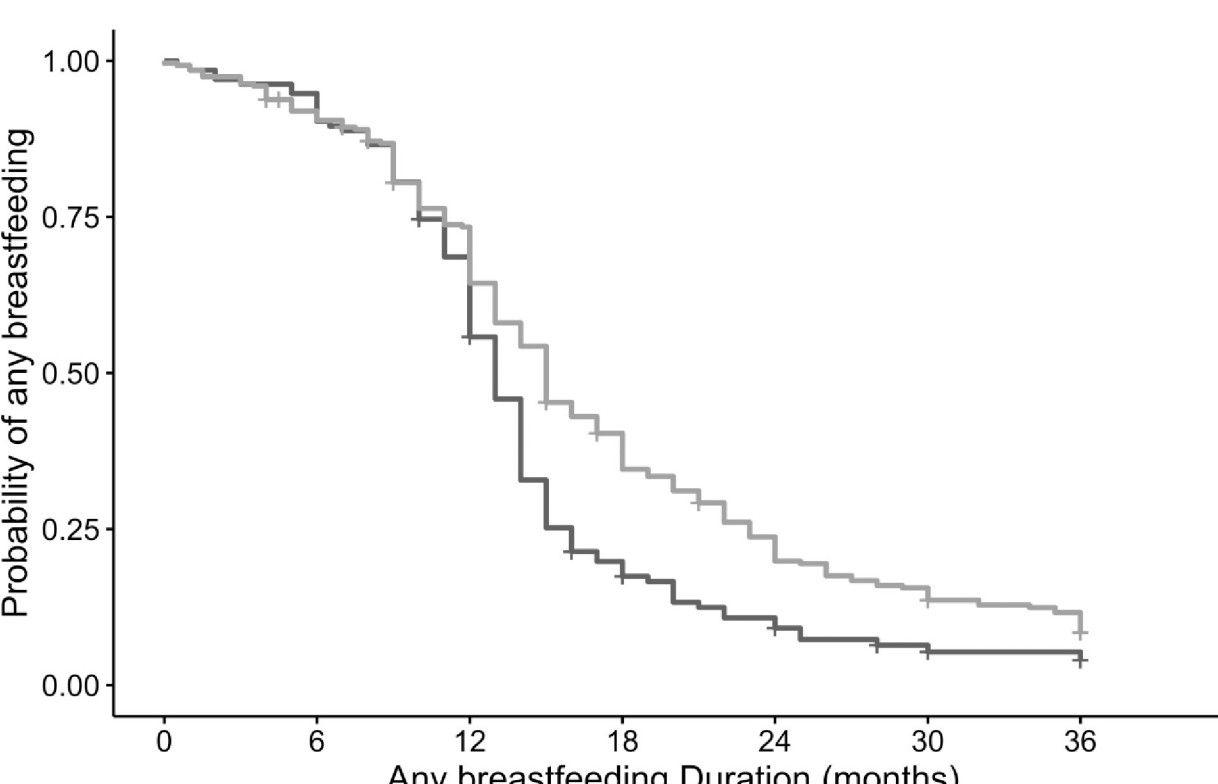

**Fig 2. Kaplan-Meier plot for any breastfeeding duration.** Kaplan-Meier survival curves for duration of any breastfeeding (in months) among participants who ever bedshared, and participants who never bedshared.

applicable to them; or if they felt they were putting their infant at risk by bedsharing but did it regardless of safety concerns. In a qualitative study of pediatricians, authors noted more than 50% of interviewees believed that bedsharing is acceptable to facilitate breastfeeding and can be practiced safely [26].

Most of the hazardous risks for sleep-related death and bedsharing are modifiable, except for preterm or low birthweight infants. It is unclear how many of the never-bedsharers had

**Table 3. Association between breastfeeding and bedsharing (Main analyses).**

| | Duration of any breastfeeding by months, mean (SD) | Univariate Test | Unadjusted HR[1] (95% CI) | p value | Adjusted HR (95% CI) | p value |
|---|---|---|---|---|---|---|
| n = 408 | **16.77 (11.16)** | | n = 408 | | n = 408 | |
| **No Bedsharing (n = 134)** | 14.08 (7.38) | log-rank test: P < 0.001 | reference | | reference | |
| **Any Bedsharing (n = 274)** | 18.08 (12.4) | | 0.66 (0.53, 0.83) | <0.001 | 0.56 (0.45, 0.71) | <0.001 |

Participants with missing any breastfeeding data were excluded and those who were still breastfeeding were censored.

*Multiple imputation was applied to impute missing values in covariates

[1]Hazard ratio: univariate cox regression was implemented to calculate the unadjusted hazard ratio for bedsharing. Multivariate cox regression was implemented to calculate the adjusted hazard ratio for bedsharing.

Table 4. Association between breastfeeding and bedsharing (Sensitivity analyses).

| | Duration of any breastfeeding by months, mean (SD) | Univariate Test | Unadjusted HR[1] (95% CI) | p value | Adjusted HR (95% CI) | p value |
|---|---|---|---|---|---|---|
| n = 522[#] | 14.48 (10.83) | | n = 522 | | n = 522[*] | |
| No Bedsharing (n = 134) | 12.45(7.52) | log-rank test: P < 0.001 | reference | | reference | |
| Any Bedsharing (n = 274) | 15.43 (11.96) | | 0.66 (0.53, 0.82) | <0.001 | 0.56 (0.44, 0.70) | <0.001 |

[#]For participants who did not fill in duration of any breastfeeding, the duration of any breastfeeding was imputed to be the reported exclusive breastfeeding duration. The imputation of duration of any breastfeeding was based on assumption that cannot be tested. Participants who were still breastfeeding and whose duration of any breastfeeding was imputed were censored.

* Multiple imputation was applied to impute missing values in covariates.

[1]Hazard ratio: univariate cox regression was implemented to calculate the unadjusted hazard ratio for bedsharing. Multivariate cox regression was implemented to calculate the adjusted hazard ratio for bedsharing.

unmodifiable circumstances that would have made bedsharing particularly unsafe in our sample of predominantly breastfeeding physicians. Future studies could explore bedsharing behaviors among non-breastfeeding physicians.

The 2022 AAP Policy states, "...we are unable to recommend bed sharing under any circumstance," and the AAP also "understands and respects that many parents choose to routinely bed share for a variety of reasons, including facilitation of breastfeeding, cultural

Table 5. Reasons respondents gave for or for not bedsharing (n = 546).

| Reasons for bedsharing | n = 367[*] |
|---|---|
| To breastfeed | 268 (73%) |
| Sleeping with my baby helped the baby or me sleep better | 215 (59%) |
| To comfort when fussy | 164 (45%) |
| To be close or bond | 104 (28%) |
| To comfort when sick | 61 (17%) |
| Not applicable | 31 (8%) |
| I thought it was safer | 28 (8%) |
| It was commonly done in my family | 27 (7%) |
| Other | 14 (4%) |
| To bottle feed | 8 (2%) |
| A doctor or nurse advised sleeping with my baby to breastfeed | 5 (1%) |
| To help with a blocked milk duct or other BF problem | 4 (1%) |
| **Reasons for not bedsharing** for not bedsharing | **n = 165[*]** |
| I thought it was safer if my baby did not sleep with me | 151 (92%) |
| A doctor or nurse advised not sleeping with my baby | 36 (22%) |
| I thought it would be too hard to get my baby to sleep in a crib when older | 23 (14%) |
| It was not commonly done in my family | 22 (13%) |
| We woke each other up, or baby woke me or others in the bed | 21 (13%) |
| Not applicable | 4 (2.4%) |
| Other | 2 (1.2%) |
| I smoke, take sedative medicine or other reason | 1 (0.6%) |

*There were 6 respondents who bedshared who did not select any reasons and 8 respondents who never bedshared and did not select reasons for not bedsharing.

preferences, and a belief that it is better and safer for their infant" [6]. While the AAP encourages clinicians to have open and nonjudgmental conversations with parents about the relative risks and benefits of bedsharing [6, 27], doing so may be challenging in the face of a strong recommendation against any bedsharing [28] and possible severe social consequences for doing so.

Growing evidence suggests proximate mother-infant sleep with breastfeeding is the evolutionary norm because it is a biological imperative, based on the physiology of human lactation [29, 30]. Bedsharing breastfed infants do not naturally sleep prone, as they roll onto their backs after feeding [31]. Additionally, bedsharing supports better maternal rest, more infant arousals, and maternal responsivity to infant arousals which may be protective [32, 33].

In our study, sleep was the second most cited reason for bedsharing. In a laboratory setting of breastfeeding mothers, 94% of routinely bedsharing mothers reported that they slept "enough" after a night of bedsharing, versus 80% of routinely solitary sleeping mothers after a night of solitary sleep [32]. Therefore, it is not surprising that breastfeeding mothers, including breastfeeding physicians, report sleep as a reason for bedsharing.

Multiple studies [1–3, 34] show a strong association between bedsharing and breastfeeding duration, the additional evidence on bedsharing and increased frequency and length of feeds with bedsharing compared to solitary sleep may suggest a causal relationship between bedsharing and breastfeeding duration [34]. Duration of breastfeeding is associated with decreased risk of other childhood diseases [25] such as obesity and leukemia, and lifetime duration of breastfeeding is associated with reduced risk of maternal disease [25] including breast cancer, ovarian cancer, myocardial infarction, hypertension, type 2 diabetes, and stroke. Bedsharing recommendations intended to reduce sleep-related deaths may inadvertently increase the risk of morbidity and mortality from other diseases in children and mothers.

Our study has multiple strengths and limitations. This survey included a large sampling with over 400 respondents for our main analysis, who had exceptionally high rates of breastfeeding, so we were able to analyze associations between bedsharing and continuation of breastfeeding. The study was not intended to be a random sampling of physician mothers; rather we aimed to examine breastfeeding physicians about their sleep practices. This intentionally selective sample of physician mothers who breastfed and were likely exceptionally motivated to breastfeed (based on average breastfeeding duration longer than national average) [35]. Selection bias about bedsharing was minimized by the vague language in the recruitment materials. Given that nearly all subjects intended to breastfeed, yet a sizable minority chose not to bedshare, confounding by breastfeeding intention is unlikely to explain the longer duration of breastfeeding among bedsharers.

One limitation is that physicians were at the most strenuous point in their careers, which could limit the generalizability to physician mothers who may have had more time or support. Some mothers at this time in their career may have spent time with their infants mostly at night due to training or career demands. Choosing the most strenuous time may have biased this sample towards decreased breastfeeding duration as well as increased bedsharing.

We did not collect data regarding maternal age, which can affect feeding choice and lactogenesis [36]; however, as nearly all respondents breastfed, this would not have been expected to have made a difference in our results. We did not ask respondents' gender identity, but gender-affirming treatments for transmen would have biased our results toward the null. Also, we did not inquire regarding the presence or absence of sleep hazards. Although 52% of bedsharers did not inform their pediatric care provider of this practice, we did not explore the reasons for this non-disclosure (i.e., were they not asked, did they not agree with bedsharing recommendations, or did they fear stigma?). We did not ask about unintentional versus

routine bedsharing or gather information about respondents' beliefs about the AAP guidelines. These questions would be informative topics for future research.

Further, one-third of respondents did not respond to questions about bedsharing or breastfeeding and these respondents were excluded. We do not know if missing bedsharing and breastfeeding data were missing at random or not, and whether this would have biased our outcomes. In addition, 114 reported duration of exclusive breastfeeding but did not report duration of any breastfeeding; however, we performed a sensitivity analysis in which we assessed this missingness of any breastfeeding duration and did not find an effect on the results. Other limitations include a cross-sectional, and retrospective collection of self-reported data and the possibility of residual confounding (unmeasured shared common cause of both bedsharing and breastfeeding e.g., personal beliefs and motivations).

## Conclusion

Despite current US recommendations against bedsharing, this practice is common among our sample of mainly breastfeeding physicians, including those who care for pregnant people and/ or infants. It is also associated with a longer duration of breastfeeding, which has implications for population health. The practice of bedsharing implies cognitive dissidence and may affect how physicians counsel about bedsharing and the AAP recommendations. Additionally, lack of disclosure of bedsharing practices has implications for practical guidance about having open non-judgmental conversations. This lack of disclosure and openness may also be a missed opportunity to counsel on bedsharing safety if most parents are practicing it.

## Supporting information

**S1 Fig. STROBE statement—Checklist of items that should be included in reports of observational studies.**
(DOCX)

**S1 Appendix. Physician mothers/birthing people infant feeding and sleep survey.**
(DOCX)

**S2 Appendix. Cox proportional hazards modeling.**
(DOCX)

**S1 Dataset. Survey dataset.**
(XLSX)

## Acknowledgments

We are grateful to Giovanna Cruz for her statistical support. We are also grateful to Drs. Andrea Braden, Kristina Lehman, and Laurie Jones for their assistance with survey distribution.

## Author Contributions

**Conceptualization:** Adetola F. Louis-Jacques, Melissa Bartick, Adeola Awomolo, Stephanie A. Leonard, Joan Meek, Susan Crowe.

**Data curation:** Adeola Awomolo, Jiaqi Zhang.

**Formal analysis:** Jiaqi Zhang, Stephanie A. Leonard.

**Investigation:** Adetola F. Louis-Jacques, Adeola Awomolo, Jiaqi Zhang, Stephanie A. Leonard, Susan Crowe.

**Methodology:** Adetola F. Louis-Jacques, Melissa Bartick, Adeola Awomolo, Jiaqi Zhang, Lori Feldman-Winter, Joan Meek, Katrina B. Mitchell, Susan Crowe.

**Project administration:** Adetola F. Louis-Jacques, Susan Crowe.

**Resources:** Adeola Awomolo, Susan Crowe.

**Supervision:** Lori Feldman-Winter, Susan Crowe.

**Validation:** Lori Feldman-Winter.

**Visualization:** Jiaqi Zhang.

**Writing – original draft:** Adetola F. Louis-Jacques, Melissa Bartick, Susan Crowe.

**Writing – review & editing:** Adetola F. Louis-Jacques, Melissa Bartick, Adeola Awomolo, Jiaqi Zhang, Lori Feldman-Winter, Stephanie A. Leonard, Joan Meek, Katrina B. Mitchell, Susan Crowe.

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
