## [Decision Letter · Decision Letter 0]

17 Jul 2023

PONE-D-23-16782Bedsharing among breastfeeding physicians: Results of a nationwide surveyPLOS ONE

Dear Dr. Louis‐Jacques,

Thank you for submitting your manuscript to PLOS ONE. After careful consideration, we feel that it has merit but does not fully meet PLOS ONE’s publication criteria as it currently stands. Therefore, we invite you to submit a revised version of the manuscript that addresses the points raised during the review process.

We look forward to receiving your revised manuscript.

Kind regards,

Linglin Xie

Academic Editor

PLOS ONE

4. Please ensure that you refer to Figure 3 in your text as, if accepted, production will need this reference to link the reader to the figure.

Reviewers' comments:

Reviewer's Responses to Questions

**Comments to the Author**

1. Is the manuscript technically sound, and do the data support the conclusions?

Reviewer #1: Yes

2. Has the statistical analysis been performed appropriately and rigorously? 

Reviewer #1: No

3. Have the authors made all data underlying the findings in their manuscript fully available?

Reviewer #1: Yes

4. Is the manuscript presented in an intelligible fashion and written in standard English?

Reviewer #1: Yes

5. Review Comments to the Author

Reviewer #1: Dear Authors,

you are come up with interesting title which creates a dilemma between d/t scholars. because some argue that bedsharing is important while others not. Additionally, your write up is smart however your paper have the following gaps:_

1.Your summary doesn't contain introduction part which is not appropriate way

2.your objective is not SMART which is not measurable .The verb you use not measurable.

3.You only use 3 kye words which is below standard.

4.Your methodology didn't go with your objective because your objective is to understand and to investigate the reason why bedsharing is high or low which is qualitative data. your study design didn't go with survival model. where is your follow up time?? Also your sample size determination and procedure is not clear?? how you select 806 physician respondents or if you incorporate all physicians is there only 806 physicians in USA?? Think over?? Why you include only single birth only?

5.result: you didn't make model fitness test, multicollinearity test and other important components were not performed

6 . what will be your interpretation of your outcome variable is it low or high requires justification??

6. PLOS authors have the option to publish the peer review history of their article (what does this mean?). If published, this will include your full peer review and any attached files.

Reviewer #1: **Yes: **Tamiru Alene,department of pediatrics and child health nursing,injibara univesity

---

## [Author Response · Author response to Decision Letter 0]

19 Feb 2024

January 15, 2024

Manuscript Number: PONE-D-23-16782

Manuscript Title: Bedsharing among breastfeeding physicians: Results of a nationwide survey

PLOS ONE

Dear Dr. Linglin Xie,

On behalf of my co-authors and myself, I would like to thank you and our reviewer for the prompt and thoughtful feedback on our manuscript. Attached, please find original comments and our responses. We are truly sorry for the delay in our response.

Reviewer #1: Dear Authors, you are come up with interesting title which creates a dilemma between d/t scholars. because some argue that bedsharing is important while others not. Additionally, your write up is smart however your paper have the following gaps:

1. Your summary doesn't contain introduction part which is not appropriate way

Authors Response: Thank you very much. We have included an introduction in the abstract.

2. your objective is not SMART which is not measurable. The verb you use not measurable.

Authors Response: The objective in the abstract has been modified to be measurable. Thank you for this comment.

3. You only use 3 kye words which is below standard.

Authors Response: Thank you, we have increased the number of key words to 5 - breast feeding, lactation, sleeping habits, postpartum period, physicians.

4. Methodology

a. Your methodology didn't go with your objective because your objective is to understand and to investigate the reason why bedsharing is high or low which is qualitative data. your study design didn't go with survival model. where is your follow up time??

Authors Response: Studying the association between bedsharing and breastfeeding duration was one of the objectives. This outcome was a time-to-event variable with right censoring, so we elected to conduct time-to-event (survival) analysis. The follow-up time was the time from birth to the end of breastfeeding. Some participants were still breastfeeding when they took the survey, and these participants were censored. I can see why using a KM curve seems odd given that we did not follow a cohort of individuals, but rather surveyed responses at one time to determine "survival of breastfeeding" given bedsharing practice. The cross-sectional survey allowed retrospective data collection of time for a survival analysis. 

The reasons for bedsharing were collected in the survey not as qualitative data. This was a secondary objective and was reported in table 5 to better understand the reasons for bedsharing or not bedsharing.

b. Also, your sample size determination and procedure is not clear?? how you select 806 physician respondents or if you incorporate all physicians is there only 806 physicians in USA?? Think over?? 

Authors Response: This is a cross sectional study design using a convenience sample of physicians who responded to the invitation to survey. We recruited as many participants as possible through 2 closed Facebook groups (Dr. MILK and Physician Moms Group) and Academy of Breastfeeding Medicine listserv. Recruitment was through social media platforms and by email listservs targeting breastfeeding physicians from October 2020 through July 2021. 

c. Why you include only single birth only?

Authors Response: Multiples were excluded since that may confound the probability of breastfeeding continuation. 

5.result: you didn't make model fitness test, multicollinearity test and other important components were not performed

We used Cox proportional hazards modeling in analyses. It is recommended that most people in a time-to-event analysis have the event observed, and in this study 73.1% had ceased breastfeeding by the time of the survey. Cox proportional hazards models are semi-parametric models that are widely used for time-to-event analyses because they do not require modeling the baseline hazard. The primary assumption in Cox proportional hazards model is that the hazard is proportional between the two comparison groups over time. The proportionality assumption was tested by the Schoenfeld residual test with global p = 0.061 on complete data, and Schoenfeld residual scaled plot. Both results showed the proportionality assumption was not violated. We also assessed the proportionality assumption for each covariate and found that it was not violated."

Table 1 Schoenfeld residual individual test result

variable Individual p value

bedsharing 0.596

race 0.861

ethnicity 0.955

Marital status 0.763

Trainee status 0.619

specialty 0.090

Birth year 2005 0.051

Depression 0.115

Figure 1 Scaled Schoenfeld residual plot (please see attached response document)

We adjusted for medical specialty, trainee status, race, Hispanic ethnicity, self-reported postpartum depression, and infant birth year before or after 2005 in the Cox proportional hazards regression. We did not have reason to suspect multicollinearity between these confounders and these were categorical or binary variables. Cox proportional hazards models are not ordinary least squares models and the confounders being categorical precludes a calculation of variation inflation factors as quantitative assessments of collinearity test.

6 . what will be your interpretation of your outcome variable is it low or high requires justification??

Authors Response: Thank you very much for this comment. We have expanded on this in the first paragraph of our discussion section.

Lines 249-253: “We found bedsharing was common among breastfeeding physicians consistent with our hypothesis, including physicians in specialties caring for women/mothers and/or infants. Also consistent with previous literature in the general population and our hypothesis, bedsharing is associated with longer duration of breastfeeding among physicians. There was a greater than 40% HR decrease for breastfeeding cessation.”

One would expect physicians to be among the most compliant groups to follow the AAP guidelines, given their type of work in the healthcare sector, and the fact that many are expected to counsel on this guideline as part of their job. This finding demonstrates that the recommendations are not consistent with the lived experiences of the physicians in our sample.

---

## [Decision Letter · Decision Letter 1]

4 Jun 2024

Bedsharing among breastfeeding physicians: Results of a nationwide survey

PONE-D-23-16782R1

Dear Dr. Louis-Jacques,

We’re pleased to inform you that your manuscript has been judged scientifically suitable for publication and will be formally accepted for publication once it meets all outstanding technical requirements.

Kind regards,

Linglin Xie

Academic Editor

PLOS ONE

Additional Editor Comments (optional):

Reviewers' comments:

Reviewer's Responses to Questions

**Comments to the Author**

1. If the authors have adequately addressed your comments raised in a previous round of review and you feel that this manuscript is now acceptable for publication, you may indicate that here to bypass the “Comments to the Author” section, enter your conflict of interest statement in the “Confidential to Editor” section, and submit your "Accept" recommendation.

Reviewer #2: All comments have been addressed

2. Is the manuscript technically sound, and do the data support the conclusions?

Reviewer #2: Yes

3. Has the statistical analysis been performed appropriately and rigorously? 

Reviewer #2: Yes

4. Have the authors made all data underlying the findings in their manuscript fully available?

Reviewer #2: Yes

5. Is the manuscript presented in an intelligible fashion and written in standard English?

Reviewer #2: Yes

6. Review Comments to the Author

Reviewer #2: (No Response)

7. PLOS authors have the option to publish the peer review history of their article (what does this mean?). If published, this will include your full peer review and any attached files.

Reviewer #2: No

---

## [Editor Report · Acceptance letter]

23 Jul 2024

PONE-D-23-16782R1 

PLOS ONE

Dear Dr. Louis‐Jacques, 

I'm pleased to inform you that your manuscript has been deemed suitable for publication in PLOS ONE. Congratulations! Your manuscript is now being handed over to our production team.

Kind regards, 

on behalf of

Dr. Linglin Xie 

Academic Editor

PLOS ONE